# Early Adolescence Prefrontal Cortex Alterations in Female Rats Lacking Dopamine Transporter

**DOI:** 10.3390/biomedicines9020157

**Published:** 2021-02-05

**Authors:** Placido Illiano, Damiana Leo, Raul R. Gainetdinov, Marta Pardo

**Affiliations:** 1The Miami Project to Cure Paralysis, Department of Neurological Surgery, University of Miami Miller School of Medicine, Miami, FL 33136, USA; 2Department of Neurosciences, University of Mons, 7000 Walloon Region, Belgium; damiana.leo@umons.ac.be; 3Institute of Translational Biomedicine, St. Petersburg State University, Universitetskaya Emb. 7–9, 199034 St. Petersburg, Russia; gainetdinov.raul@gmail.com; 4St. Petersburg University Hospital, St. Petersburg State University, Universitetskaya Emb. 7–9, 199034 St. Petersburg, Russia; 5Department of Neurology and Molecular and Cellular Pharmacology, University of Miami Miller School of Medicine, Miami, FL 33136, USA

**Keywords:** dopamine transporter, knockout, prefrontal cortex, neurodegeneration, neuroinflammation, adolescence

## Abstract

Monoamine dysfunctions in the prefrontal cortex (PFC) can contribute to diverse neuropsychiatric disorders, including ADHD, bipolar disorder, PTSD and depression. Disrupted dopamine (DA) homeostasis, and more specifically dopamine transporter (DAT) alterations, have been reported in a variety of psychiatric and neurodegenerative disorders. Recent studies using female adult rats heterozygous (DAT+/−) and homozygous (DAT−/−) for DAT gene, showed the utility of those rats in the study of PTSD and ADHD. Currently, a gap in the knowledge of these disorders affecting adolescent females still represents a major limit for the development of appropriate treatments. The present work focuses on the characterization of the PFC function under conditions of heterozygous and homozygous ablation of DAT during early adolescence based on the known implication of DAT and PFC DA in psychopathology during adolescence. We report herein that genetic ablation of DAT in the early adolescent PFC of female rats leads to changes in neuronal and glial cell homeostasis. In brief, we observed a concurrent hyperactive phenotype, accompanied by PFC alterations in glutamatergic neurotransmission, signs of neurodegeneration and glial activation in DAT-ablated rats. The present study provides further understanding of underlying neuroinflammatory pathological processes that occur in DAT-ablated female rats, what can provide novel investigational approaches in human diseases.

## 1. Introduction

Genetically determined changes in dopaminergic (DA) activity might contribute to the occurrence of neuropsychiatric diseases such as PTSD [1,2] and ADHD [3]. Sex-specific studies have demonstrated that the prevalence of negative outcomes associated with childhood ADHD [4] and PTSD [5] are higher in women, as well as the incidence of PTSD [6]. However, little research has focused on sex-specific studies of these pathologies at both clinical [4,6] and preclinical level [7,8]. Recent studies focusing on adult dopamine transporter deletion in rats, in the heterozygous and homozygote genotypes, highlighted that female adult rats from both genotypes display model validity for PTSD and ADHD [9]. On the other hand, male adult DAT−/− rats also display model validity for ADHD [10,11]. Furthermore, Adinolfi and coworkers have shown that adolescent DAT−/− rats display cognitive alterations [12]. In line with our previous studies, DAT−/− female rats provide a highly suitable animal model to investigate the less known pathophysiology of disorders involving aberrant DA function related to DAT ablation affecting females.

DA homeostasis has been shown to play an important role in prefrontal cortex (PFC) neuromodulation during postnatal development [13]. Monoamine dysfunctions in PFC are central features of psychiatric and neurodegenerative disorders [14]. It is well established that occurrence of a traumatic event during adolescence [4,15], as well as developmental changes that occur during that stage [4], might increase the risk for PTSD and ADHD, where DA homeostasis has been shown to be altered [16,17]. Enhanced inflammatory conditions [18] and neurodegenerative hallmarks related to DAT alterations [19,20,21] are among the most relevant underlying mechanisms of aberrant PFC function in these disorders. Recent studies have also shown the important role played by astrocytes in the control of DA homeostasis in the PFC during early adolescence [13,19], when glutamatergic synapses in this area are at an important maturational stage [20,21,22].

In this framework, we sought to investigate the changes that occur in the PFC of female DAT+/− and DAT−/− rats, with a focus on early adolescence. We aimed to provide a more comprehensive understanding of the role of DAT dysfunction and its impact on glial homoeostasis to determine its role in the pathophysiology of several psychiatric and neurodegenerative diseases. Furthermore, we also investigated parts of the glutamatergic signaling due its known interaction with the DA system [23,24,25], and dysfunctions described in a variety of psychiatric disorders [26,27,28] with high prevalence in women and onset during adolescence.

This work focused on the PFC, for its role in controlling executive function during early adolescence, integrating investigations on the mechanism of neurodegenerative diseases that involve glial cell function with the use of this new rat mutant model of aberrant DA neurotransmission. Herein, we report that genetic ablation of DAT in the adolescent PFC leads to hyperactive phenotype, altered glutamatergic neurotransmission, neurodegeneration and glial cell activation in DAT+/− and DAT−/− rats.

## 2. Methods

### 2.1. Animals

Wistar Han heterozygous (DAT+/−) and homozygous (DAT−/−) rats were generated as previously described [29]. Wildtype (DAT+/+), DAT+/− and DAT−/− female rats were weaned at post-natal day (PND) 21 and housed in groups of 2–3, with food and water available ad libitum. The colony was maintained under standard conditions (12 h light/dark cycle 6 a.m. on–6 p.m. off, 21 ± 1 °C, 40–70% humidity). The breeding scheme was performed as reported from our group, mating mature DAT+/− female rats of fertile age (>2 months old) with mature DAT+/− male rats [9,30]. We acknowledge the possible effect that the breeding strategy could have on the study [30], out of our scope of work. All of the experiments were conducted in accordance with the National Institute of Health Guide for Care and Use of Laboratory Animals and approved by the University of Miami IACUC (Protocol #17-016, Approval Date 15 June 2017). The replacement, reduction and refinement (3R) principle was applied, and the total number of animals used was 45.

### 2.2. Locomotor Activity

Locomotor activity was evaluated using Omnitech Digiscan apparatus (AccuScan Instruments, Columbus, OH, USA) under illuminated conditions. The plexiglass apparatus (40.4 × 40.5 × 30.3 cm chamber) was equipped with four open field monitors, each made of 16 light beams placed on the horizontal x and y axes. The hardware detected beam breaks, while the software determined the location of the animal in the locomotor box apparatus. The horizontal activity was expressed in beam cross number in the x axes, in counts. Vertical activity represented the number of beam breaks on the y axes, in counts, which originated from rearing activity of the rat. Stereotyped behavior (stereotypies count), which measures the beam break patterns occurring in time intervals lower than 1 s, was also provided in counts.

The animals were individually tested for 85 min at the age of PND35. The number of rats per group was as follows: DAT+/+ *n* = 14, DAT+/− *n* = 18, DAT−/− *n* = 13.

### 2.3. Western Blotting

Rats were sacrificed by decapitation following locomotor activity recording. Half brain hemispheres were sectioned, snap-frozen and kept at −80 °C until analysis.

Western blotting experiments were performed, as previously described [9], with minor modifications. Prelimbic PFC tissue was dissected from one hemisphere and mechanically homogenized in extraction buffer (20 mmol/L Tris–HCl, pH 7.4, 150 mmol/L NaCl, 1% Triton-X-100, 1 mmol/L EDTA, 1 mmol/L EGTA) supplemented with PhosSTOP™ (Millipore Sigma #4906837001) and Complete Mini, EDTA-free (Roche, 11836170001). Samples were left on ice for 20 min and then tip-sonicated for 3 s at medium speed. Samples were centrifuged for 20 min at 4 °C and 10,000× *g*. Protein concentration was measured using a Bradford assay (Bio Rad #500-0205). The protein extracts (60 μg) were run on 4–20% Criterion TGX Precast Midi Protein Gel (Bio Rad #5671094) and transferred to nitrocellulose membranes (Bio Rad #1620112). The blots were blocked in 5% nonfat dry milk (RPI International #M17200-500.0) in PBS-Tween 20 0.05% and incubated overnight at 4 °C with the following primary antibodies: NMDAR1 (Millipore #AB9684; 1:1000); NMDAR2B (Millipore #06-600; 1:1000); PSD95 (Abcam #12093; 1:1000); VAMP2 (Millipore #AB3347, 1:1000); Caspase 3 (Cell Signaling #9662; 1:500); ALDH1L1 (Abcam #87117; 1:1000); MBP (MAB #386; 1:1000); phosphoP38(T180/Y182) (Cell Signaling #9216; 1:500); total P38 (Cell Signaling #8690; 1:500); CD45 (eBioscience #14-0451-82; 1:500); β-Actin (Sigma #A2228, 1:2500); and β-tubulin (Sigma #T8535, 1:10000). Where needed, the membranes were stripped for 10 min with Restore™ PLUS Western Blot stripping buffer (Thermo Scientific #46430), blocked and incubated with primary antibody, as described above. After washing in PBS-Tween 20 0.05%, the membranes were incubated for 1 hour at room temperature with the appropriate horseradish peroxidase (HRP) conjugated secondary antibody in PBS-Tween 20 0.05%. Following secondary antibody incubation, the membranes were washed and incubated with ECL detection reagent (Thermo Fisher #34578) for 1–2 min. Immunoblots were acquired using the Bio Rad ChemiDoc™ system, and images were quantified using Image Lab 5.2.1 software. Depending on the size of each protein, bands were normalized for housekeeping gene (β-actin, tubulin) or for total form of related protein (Caspase 3, P38) and expressed as relative to control (DAT+/+). The number of rats per group was as follows: DAT+/+ *n* = 6, DAT+/− *n* = 6, DAT−/− *n* = 5.

### 2.4. Tissue Preparation for Histological Assessments

Rats were sacrificed as indicated in the “Western blotting” section. The remaining half-brain was post-fixed in paraformaldehyde 4% in PBS for 3 days at 4 °C, then transferred in sucrose 0.5 M in PBS 0.1 M for 7 days at 4 °C for cryoprotection, frozen on dry-ice surface and kept at −80 °C until analysis.

Coronal serial sections were collected on a freezing microtome (Leica), and coordinates followed were +2.52  to −1.44 mm from bregma (antero-posteriorly) [31], in 15 series at a thickness of 30 μm. In order to remove autofluorescence, slices were quenched in sodium borohydride (0.5% in PBS) for 10 min and rinsed 3 times with PBS. Slices were permeabilized with PBS-Triton-X 0.4% and blocked in 5% NGS in 0.4% Triton-X at room temperature for 1 h. Immunohistochemical staining was performed using antibodies raised against NeuN (Millipore #MAB377; 1:500) and Iba1 (Wako #019-19741; 1:500) in 5% NGS in 0.4% Triton-X and maintained overnight at 4 °C. Slices were rinsed 3 times with PBS-Triton-X 0.1% and then incubated with secondary antibody, Alexa 594 or Alexa 488 (1:750 in PBS/0.1% Triton-X) for 2 h at room temperature, followed by 3 washes in PBS-Triton-X 0.1%. Slices were stained with DAPI (4′,6-diamidino-2-phenylindole Invitrogen D1306, 1:6000 in PBS-Triton-X 0.1%) for 1 min prior to mounting with Fluoromount aqueous medium (Sigma-Aldrich F4680).

Fluoro-Jade C (FJC) staining was performed on the same PFC slice series following manufacturer’s instructions (TR-100-FJT). Briefly, slides were dried at 60 °C on a slide warmer for 10 min and incubated in a mixture of 80% ethanol/Solution A (9:1) for 5 min. Slides were then rinsed 2 times in ethanol 70% and 2 times in double-distilled (dd) water. Slides were incubated in a mix of dd water/Solution B (9:1) for 10 min and rinsed in dd water. A mixture of FJC staining solution, DAPI and dd water (1:1:8) was added to the slides and incubated in the dark for 10 min; after 2 rinses in dd water, slides were dried at 60 °C on a slide warmer for 10 min and then cleared in xylene. Slides were mounted in DPX mounting medium.

### 2.5. Image Analyses

In the case of NeuN^+^ cell (neuronal) count, data were analyzed using FIJI software [32]. Briefly, 3–4 images acquired at 10× were obtained for 4 animals per genotype (pixel conversion: 1 pixel ≡ 1.024 μm). After conversion to 8-bit, images were set to a threshold lower than 8% and cell bodies counted using the Analyze > analyze particles function. Experimenters were blinded to the genotype conditions.

Morphometric analyses of Iba1^+^ microglia were performed as previously reported [33], with minor modifications. In summary, data were analyzed using FIJI software [32], and 3 image stacks acquired at 63× were obtained for 3 animals per genotype (pixel conversion: 1 pixel ≡ 0.1652 μm). Cell tracing was performed using the “Simple Neurite Tracer” plugin and Sholl analyses executed as previously reported [34]. Experimenters were blinded to the genotype conditions.

### 2.6. ELISA

PFC protein extracts prepared for Western blotting were used for ELISA analysis. Thirty μg of protein extract per sample was used. Concentrations of interferon gamma (IFNγ) were measured using commercially available enzyme-linked immunosorbent assays following manufacturer indications (Pierce biotechnologies, Rockford, IL 61105, USA, Thermo Scientific #EM10015). Each sample was analyzed in duplicate, and all the absorbance readings were performed at 450 and 550 nm. The final absorbance value was obtained subtracting the 450 nm readings from the 550nm values. Protein extract dilution was tested as per manufacturer instructions and chosen to be 30 μg to fall within the dynamic range of the calibration curve. DAT+/+ *n* = 5; DAT+/− *n* = 4; DAT−/− *n* = 5.

### 2.7. Statistical Analysis

GraphPad Prism 9 was used for all analyses, and the null hypothesis was rejected at the 0.05 level of significance. The locomotor behavior experiment results were analyzed with two-way ANOVA, followed by Tukey’s multiple comparison post hoc test. One-way ANOVA with Tukey’s post hoc test was used for analysis of Western blot and ELISA data. Nested one-way ANOVA followed by Tukey’s multiple comparison post hoc test was used to analyze cell count data and morphometric analyses.

## 3. Results

### 3.1. Locomotor Activity of Female Adolescent DAT+/− and DAT−/− Rats

At PND35, rats were tested for locomotor behavior and then sacrificed for biochemical analyses of the PFC (Figure 1A).

Ablation of DAT causes dramatic increase of the horizontal activity (Figure 1B) (two-way ANOVA F (32, 688) = 3.676; *p* < 0.0001) and of the stereotypies count (two-way ANOVA F (32, 672) = 1.739; *p* = 0.0076; Figure 1C) already in adolescent DAT−/− female rats, while vertical activity is not affected (two-way ANOVA F(32, 688) = 0.6379; *p* = 0.9410; Figure 1D). These results indicate that knockout of DAT enhanced locomotor behavior during development in female rats.

### 3.2. Glutamatergic and Synaptic Alterations in Female Adolescent DAT+/− and DAT−/− Rats

Cortical alterations of glutamate neurotransmission might contribute to the exhibition of hyperactive phenotype in rats [35]. Therefore, we investigated the expression levels of glutamate subunits in the PFC of DAT+/− and DAT−/− rats. While expression levels of the obligate NMDAR1 subunit were unchanged among the three genotypes (Figure 2A; one-way ANOVA F(2,14) = 1.383; *p* = 0.2831), a significant decrease of NMDAR2 subunit was observed in the cortex of both DAT+/− and DAT−/− rats, compared to DAT+/+ controls (Figure 2B; one-way ANOVA F(2,14) = 15.53; *p* = 0.0003; Tukey’s multiple comparison test *p* = 0.0005 DAT+/+ versus DAT+/−; *p* = 0.0015 DAT+/+ versus DAT−/−).

Further, we sought to investigate if ablation of DAT had substantial effects on the main components of pre- and post-synaptic machinery. DAT−/− rats display a significant increase of protein levels of pre-synaptic protein VAMP2 (Figure 2C; one-way ANOVA F (2, 13) = 6.051; *p* = 0.0139; Tukey’s multiple comparison test *p* = 0.0186 DAT+/+ versus DAT−/−). In order to study possible post-synaptic alterations in the PFC, we analyzed the expression of PSD95 protein. Our data indicate that PSD95 levels were unchanged among the three genotypes (Figure 2D; one-way ANOVA F (2, 14) = 0.1729; *p* = 0.8430].

These data indicate that either partial of total ablation of DAT affects glutamate receptor homeostasis in the developing PFC of DAT+/− and DAT−/−. Moreover, DAT−/− rats also show anomalies in the VAMP2 component of the pre-synaptic machinery.

### 3.3. DAT−/− Rats Display Neuronal Cell Death in the Prefrontal Cortex

In order to assess whether absence of DAT could cause neuronal cell death as previously shown in the DAT−/− mouse model in the striatum [36], we initially sought to qualitatively investigate the occurrence of neuronal death by Fluoro-Jade C (FJC) staining [37]. As shown (Figure 3A), elevated FJC staining positivity was observed in DAT−/− PFC, while a sparse signal was detected in DAT+/− and complete absence in DAT+/+ controls. Based on FJC qualitative data, we proceeded to investigate markers of apoptotic response such as cleaved caspase 3 [38]. Western blot analysis showed that levels of cleaved caspase 3 are higher in the PFC of DAT+/− rats (one-way ANOVA F (2, 13) = 5.898; *p* = 0.0150; Tukey’s multiple comparison test *p* = 0.0545 DAT+/− versus DAT+/+ controls) albeit without reaching statistical significance (Figure 3B). Nevertheless, DAT−/− rats display a 2-fold increase in the levels of cleaved caspase 3 (Tukey’s multiple comparison test *p* = 0.0184 DAT−/− versus DAT+/+ controls; Figure 3B).

In order to further address the occurrence of neuronal cell death and concomitant elevation of cleaved caspase 3 expression, we performed PFC NeuN^+^ cell body counting (Figure 3C,D). PFC tissue from early adolescent females showed a marked reduction in the numbers of NeuN^+^ cells in DAT−/− genotype (nested one-way ANOVA F (2, 9) = 10.21; *p* = 0.0048; Tukey’s multiple comparison test *p* = 0.0048 DAT+/+ versus DAT−/−; *p* = 0.0246 DAT+/− versus DAT−/−).

Taken together, our data revealed neuronal cell death in the PFC of DAT−/− rats.

### 3.4. Pro-Inflammatory Phenotype in the Prefrontal Cortex of DAT−/− Rats

We sought to investigate whether absence of DAT could also potentially impact glia in DAT+/− and DAT−/− rats. To this aim, we measured the levels of ALDH1L1, a selective pan-astrocyte marker [39,40]. DAT−/− rats showed a 1.5-fold increase in the levels of ALDH1L1 (one-way ANOVA F (2, 14) = 11.79; *p* = 0.0010; Tukey’s multiple comparison test *p* = 0.0007 DAT−/− versus DAT+/+ controls), while DAT+/− rats showed a trend toward elevation that did not reach statistical significance (Tukey’s multiple comparison test *p* = 0.0586 DAT+/− versus DAT+/+ controls; Figure 4A). Moreover, we analyzed the levels of myelin basic protein (MBP), which is responsible for axon myelination and the expression of which is tightly regulated by mature oligodendrocytes [41]. We did not observe any variation in the expression of MBP in the PFC of either DAT+/− or DAT−/− rats (one-way ANOVA F (2, 14) = 0.8489; *p* = 0.4488; Tukey’s multiple comparison test *p* = 0.9437 DAT+/+ versus DAT+/−; *p* = 0.4373 DAT+/+ versus DAT−/−). These results provide the first evidence of possible astrogliosis in the PFC of DAT−/−, which does not alter oligodendrocyte myelination.

In order to further investigate the occurrence of inflammatory signal transduction, we measured the levels of expression of mitogen-activated protein kinase (MAPK) P38, which plays a pivotal role in the regulation of inflammatory mediators and cytokines [42]. Our data show that phosphorylation levels of phospho-p38 MAPK (Thr180/Tyr182) were significantly increased in the PFC of DAT−/− rats (one-way ANOVA F (2, 14) = 8.332; *p* = 0.0041; Tukey’s multiple comparison test *p* = 0.0007 DAT−/− versus DAT+/+ controls). We then analyzed the levels of interferon gamma (IFNγ), a pro-inflammatory cytokine involved in glial activation [43], in the PFC of DAT+/− and DAT−/− rats. We observed a significant increase of IFNγ in female adolescent DAT−/− rats (one-way ANOVA F (2, 11) = 5.621; *p* = 0.0208; Tukey’s multiple comparison test p=0.0311 DAT−/− versus DAT+/+ controls; Figure 4C).

To further investigate whether ablation of DAT causes glial activation, we also analyzed the levels of expression of CD45, a marker for microglia in the CNS [44]. Levels of CD45 were unchanged among the three genotypes (one-way ANOVA F 92, 14) = 0.5094; *p* = 0.6116). In order to further assess whether microglial activation might still occur, we assessed Iba1+ microglia three-dimensional cell morphology (Figure 5 and Figure 6). Specifically, Iba1+ cells showed longer process length in DAT−/− PFC (nested one-way ANOVA F (2, 6) = 11.97; *p* = 0.0080; Tukey’s multiple comparison test *p* = 0.0099 DAT+/+ versus DAT−/−; *p* = 0.0179 DAT+/− versus DAT−/−; Figure 5B and Figure 6), albeit the maximum intersection radius did not change among genotypes (nested one-way ANOVA F (2, 6) = 1.672; *p* = 0.2647; Figure 5C). Microglia from DAT−/− also displayed a higher number of branches (nested one-way ANOVA F (2, 24) = 37.00; *p* < 0.0001; Tukey’s multiple comparison test *p* < 0.0001 DAT+/+ versus DAT−/−; *p* < 0.0001 DAT+/− versus DAT−/−; Figure 5D and Figure 6). Sholl analysis highlighted a significant interaction between genotype and number of intersections (two-way ANOVA F (98, 294) = 1.944; *p* < 0.0001; Figure 5E), where DAT−/− rat PFC microglia had higher intersection counts than DAT+/− and DAT+/+ microglia between 14 and 20 μm distance from the nucleus (Tukey’s multiple comparison test 0.03< *p* <0.0006). Albeit to a lesser extent, DAT+/− microglia also had significantly higher numbers of intersections than DAT+/+ controls at 15 μm (Tukey’s multiple comparison test *p* = 0.0163). Taken together, these data reveal that DAT ablation causes glial alterations highlighting a pro-inflammatory state in the PFC of female adolescent DAT−/− rats.

## 4. Discussion

Homozygous ablation of DAT recapitulates the main features of human ADHD in mice [45,46,47] and in rats [10,11,12,29]. Furthermore, DAT heterozygosis affects neurodevelopment in mice consistent with an ADHD phenotype [48,49], and it increases vulnerability to stress in adult female rats [9]. Recent studies on human DAT have highlighted that mutations of this protein lead to diverse neuropsychiatric diseases [50] such as ADHD [51,52], bipolar disorder [53], PTSD [2] and to neurodegenerative disorders like dopamine transporter deficiency syndrome (DTDS) [54,55].

The functional state of DAT has been proven to affect neuronal plasticity and control of cognitive function in the PFC [56,57], where pharmacological inhibition has shown to induce pro-cognitive effects [58]. DA function in the PFC controls several processes including working memory, attention, flexibility in behavior and action planning [59]. The DAT−/− rat model displays aberrant PFC signaling [29,60] that contributes to the alterations of cognitive, social and sexual behavior [10,29,60] in adult rats, with occurrence of behavioral abnormalities starting during adolescence [12]. More recently, investigations on PFC during postnatal development revealed that DA homeostasis is tightly regulated by astroglial cells at this stage [13], thus highlighting the importance that glial cells play in regulating DA signaling during development. Indeed, several studies have confirmed that DA can exert either anti-inflammatory or pro-inflammatory effects in human and in vivo models through dopamine receptor activation on astrocytes and microglia [61,62,63]. In this framework, we sought to investigate the effects of DAT ablation in the prelimbic PFC, responsible for cognitive processing [64], that could lead to glial pro-inflammatory process that might contribute to the underlying pathophysiology of female DAT+/− and DAT−/− rats during early adolescence.

Our data indicate that adolescent DAT−/− female rats display an overt hyperactive phenotype in the locomotor chamber, characterized by increased sustained locomotion as well as stereotypic behavior. Our data confirm previous findings observed in this animal model [9,10,29] and demonstrate occurrence of sustained hyperactive phenotype in adolescent female rats.

The ensemble of behavioral phenotypes observed in DAT−/− rats [10,11,12,29,30] depends primarily on the different roles that DA neurotransmission plays in the mesocortical system and in the PFC [58,65], and it is ascribable to alterations in the cross-talk with glutamatergic neurotransmission [23,24,25]. Disruption of these pathways has been widely linked to psychiatric disorders [26] including autism, schizophrenia [27] and ADHD [28]. Indeed, adult DAT−/− male rats display glutamate release alterations in the nucleus accumbens [60].

Our present data indicate that neither heterozygous nor homozygous deletion of DAT affects the expression levels of the obligate NMDA receptor subunit R1 (NMDAR1), while a reduction of NMDA receptor 2B (NMDAR2B) glutamate subunit is observed in both genotypes. NMDAR2B is the main component of neuronal extra-synaptic NMDARs [66] that guarantees the maintenance of sustained activity and proper postnatal development of the PFC [67]. Our data suggest that reduction of NMDAR2B in the PFC of female DAT−/− rats during adolescence could contribute to aberrant cortical development, which includes pre-synaptic alterations observed in the increased levels of VAMP2. Further studies investigating the mechanisms responsible for such impacts on glutamatergic signaling are needed. However, the hyperdopaminergic state of DAT−/+ as well as DAT−/− could be underlying such impact on the glutamatergic pathway components.

In neuropsychiatric diseases, extrasynaptic NMDARs are involved in complex signaling mechanisms that promote cell death, ultimately by activation of P38 MAPKs [66]. The marked increase in P38 phosphorylation that we measured in the PFC of homozygote animals might therefore underlie the neuronal cell death observed in the PFC of adolescent female DAT−/− rats. A dual approach [37] was previously used to confirm neurodegeneration in DAT−/− PFC that included (i) Fluoro-Jade C (FJC) staining, where a positive result indicates the presence of degenerating neurons [37,68], and (ii) NeuN^+^ cell bodies count, the reduction of which indicates neuronal loss. We further investigated neurodegeneration addressing the concurrent activation of apoptotic cleaved caspase 3. Ultimately, we detected no changes in the levels of glial cell type markers CD45 (microglia) and MBP (oligodendrocytes), and increased levels of ALDH1L1 astroglial marker. In conclusion, these data indicate that ablation of DAT is responsible of substantial changes that involve multiple pathways leading to neurodegeneration in the PFC of DAT−/− rats. Interestingly, a trend of increased cleaved caspase 3 and spare positivity to FJC staining was observed in female DAT+/− rats that, in combination with the reduced NMDAR2B protein levels, opens novel investigational scenarios oriented toward specifically addressing changes that could occur when a mutation is only present in one allele and DAT is only partially reduced. Furthermore, neurodegeneration is also observed in the striatum of a subpopulation of DAT−/− mice (~36%), leading to a fatal neurodegenerative phenotype reminiscent of DTDS symptomatology [36,47].

It is noteworthy that P38 MAPK module activation plays a pivotal role in inflammatory responses and cytokine release [69] and that increased inflammation in the central nervous system is a hallmark of psychiatric diseases in humans [18]. Dopamine is also an important regulator of cytokine secretion, regulating immune response [70] and neuropsychiatric disorders where aberrant DA neurotransmission is involved, such as PTSD and schizophrenia, displaying increased CNS inflammation [71]. Astrocytes are considered as initiators of and responders to inflammation in the CNS [72] and preside over the control of DA homeostasis in the developing PFC [13]. Based on our data and on previous literature, we sought to investigate glial markers indicative of pro-inflammatory signal transduction in the PFC of DAT+/− and DAT−/− adolescent rats. Homozygous ablation of DAT presented a characteristic profile of pro-inflammatory state [43] on the PFC indicated by increases in pan-astroglial marker (ALDH1L1), phosphorylation of P38 MAPK and an increase in the levels of pro-inflammatory cytokine IFNγ. Such an environment might likely affect microglia, which respond to inflammation processes in the brain and regulate neuronal activity as well as connectivity [73]. Indeed, the pro-inflammatory profile seen in the PFC of adolescent DAT−/− rats was accompanied by changes in microglial morphology consistent with neuroinflammation [74], characterized by increased arborization and complexity. Peculiarly, the absence of alteration of myelin basic protein MBP levels produced by mature oligodendrocytes suggests that the pro-inflammatory conditions present in the PFC of female DAT−/− rats might not affect this glial population.

Altogether, these data support the fact that the PFC of DAT−/− female adolescent rats presents a pro-inflammatory profile at early stages in life, concurrent with neurodegeneration. Importantly, these patterns indicative of increased inflammatory responses were not present in DAT+/− female adolescent rats. Our differential data obtained from DAT−/− and DAT+/− rats might, therefore, provide further insights in the PFC pathophysiology of neuropsychiatric diseases like ADHD and PTSD, respectively, the model validity of which was demonstrated in our previous work [9]. An increasing number of studies using DAT+/− and DAT−/− rats are underway, and based on our results and previous publications [10,12], partial mutation of DAT is sufficient to cause several alterations to a variety of underlying mechanisms controlling, among others, behavioral parameters relevant for the study of psychiatric disorders [9,10,12].

Prevalence of psychiatric and neurodegenerative disorders is determined by sex [75] and should be treated accordingly [76]. Psychiatric disorders such as ADHD [77] and PTSD present higher incidence and prevalence in women compared to men [78,79]. Although most psychiatric disorders cited across our study present higher prevalence in women, we acknowledge that our study focused only on female rats, and further studies are needed to expand our results also in male rats. Additionally, we studied female rats at PND35, the time point during development when hormones have been described to be almost ready to start the first proestrus cycle [80]. We acknowledge the need for future studies considering the role of hormones modulating glutamatergic synaptic response. The present investigation focused on early adolescent rats. We acknowledge the limitations of the time frame selected and the need to introduce additional time points during development. Future studies are needed to ensure better understanding of possible glial and neurodegenerative features during development and in adult rats.

Neuroinflammation is a common facet of a large variety of psychiatric and neurodegenerative disorders. In the present study we showed in early adolescent females that genetically engineered female rats lacking DAT present with a neurodegenerative phenotype in the PFC, accompanied by altered glutamatergic neurotransmission and glial activation at early stages of adolescence. To our knowledge, this is the first characterization of the neuroinflammatory phenotype of the PFC of female DAT KO rats during adolescence. Further studies are needed to deeply characterize neuroinflammation in our DAT+/− and DAT−/− rat model at different developmental stages, including adulthood and across sexes. Our study introduces a novel investigational perspective that may be pursued to shed light on the neuroinflammatory and neurodegenerative pathophysiology of disorders involving DAT and aberrant DA neurotransmission.

## Figures and Tables

**Figure 1 biomedicines-09-00157-f001:**
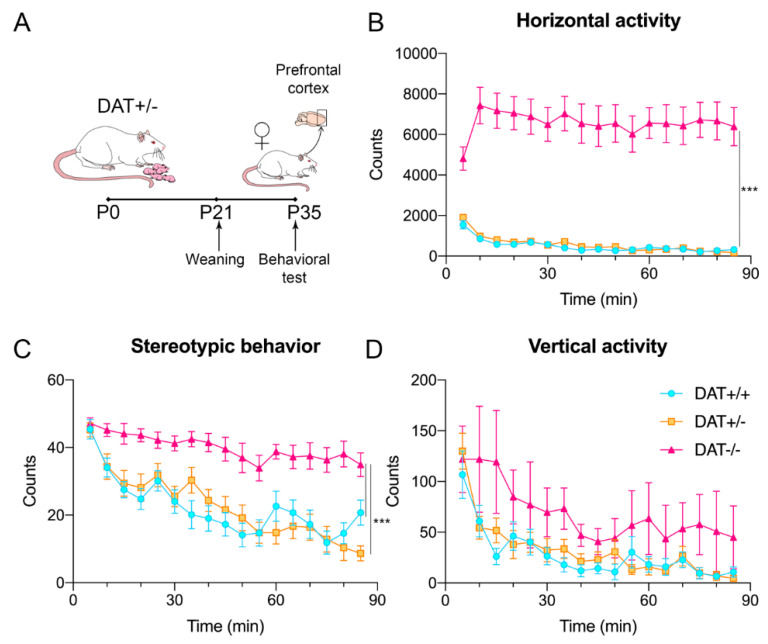
Behavioral deficits of female adolescent DAT−/− rats. (**A**) Schematic illustration of the experimental timeline. Rats were culled by DAT+/− mothers and weaned at PND21; at PND35, female rats were tested for locomotor activity. Brain tissue was isolated immediately after behavioral test. Behavioral activity in the open field: (**B**) horizontal activity, (**C**) stereotypic behavior counts and (**D**) vertical activity (**D**). Data are presented as mean ± SEM; *** *p* < 0.001.

**Figure 2 biomedicines-09-00157-f002:**
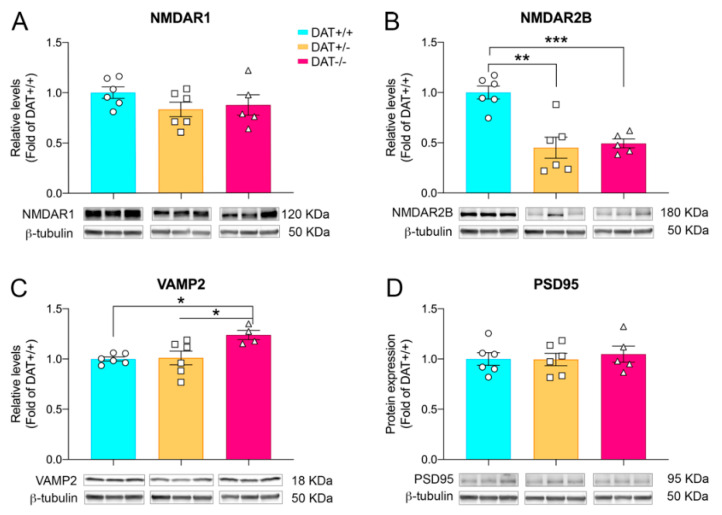
Glutamatergic alterations in female adolescent DAT+/− and DAT−/− rats. (**A**) Ablation of DAT does not affect NMDAR1 levels. (**B**) Reduction of NMDAR2B levels were found in DAT+/− and DAT−/− adolescent female rats. (**C**) Significantly increased levels of pre-synaptic VAMP2 protein were observed in DAT−/− rats. (**D**) No changes were observed for post-synaptic PSD95 protein levels. Quantification and representative bands are presented. Individual data points are shown for each group. Data are presented as mean ± SEM; * *p* < 0.05, ** *p* < 0.01, *** *p* < 0.001.

**Figure 3 biomedicines-09-00157-f003:**
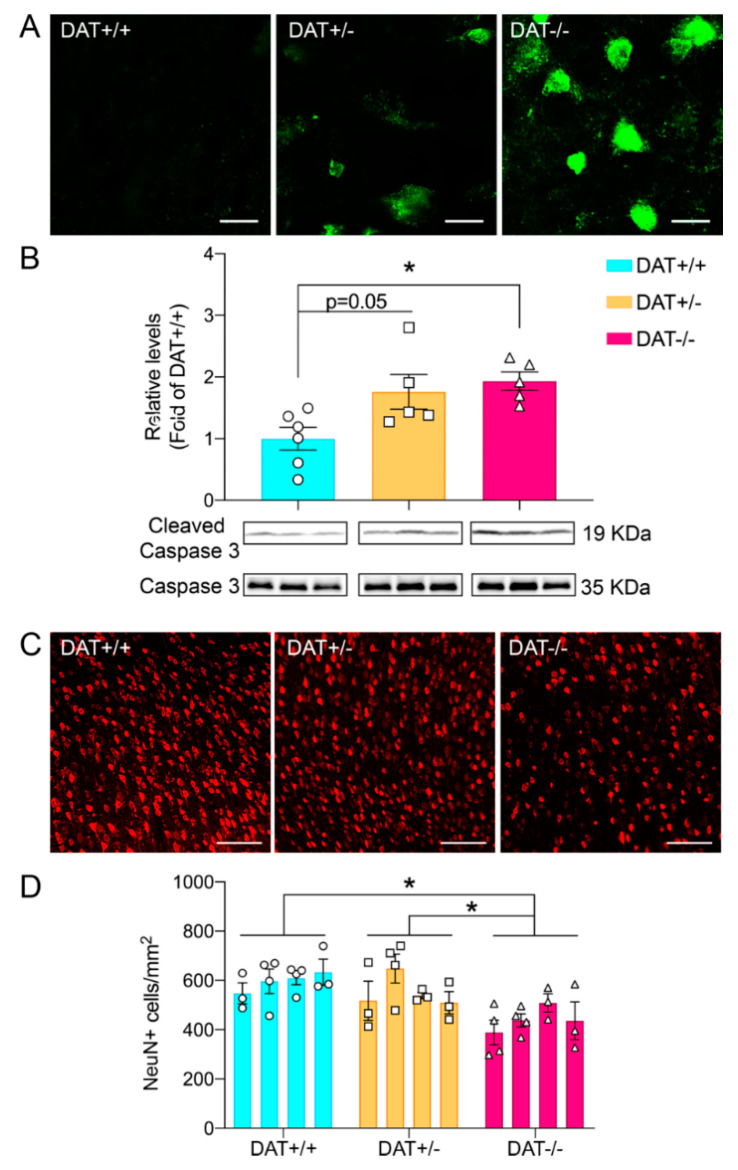
Effects of DAT ablation on neuronal cell death in the prefrontal cortex. (**A**) Fluoro-Jade staining in DAT+/− and DAT−/− rats displaying degenerating neurons in DAT−/− rats. Scale bar: 3 μm; (**B**) DAT−/− rats showed a significant increase in levels of cleaved caspase 3. (**C**) Reduction in numbers of NeuN^+^ cells in prefrontal cortex (PFC) of DAT−/− rats. Scale bar: 20 μm (**D**). Quantitation of NeuN^+^ cells in PFC. Three to four images were analyzed, n = 4 rats for each genotype; individual data points are shown. Data are presented as mean ± SEM, * *p* < 0.05.

**Figure 4 biomedicines-09-00157-f004:**
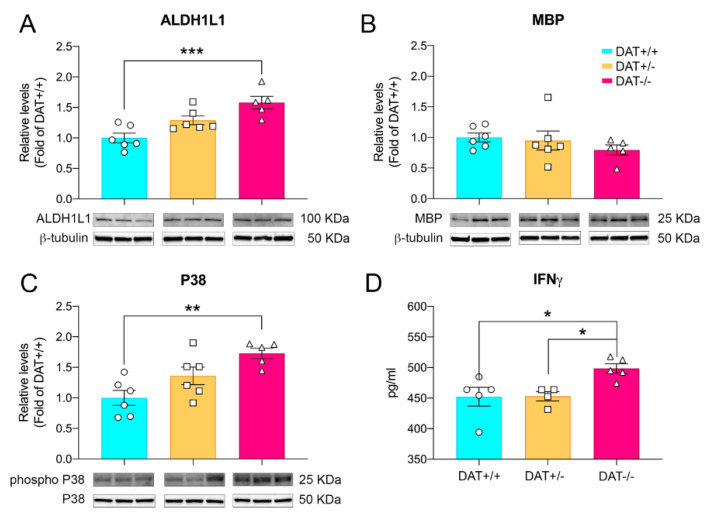
Increase in markers of neuroinflammation in DAT−/− rats. (**A**) DAT−/− rats showed an increase in levels of astroglial marker ALDH1L1. (**B**) Levels of myelin basic protein were unchanged among the three genotypes. (**C**) Phosphorylation levels of MAPKp38 were significantly increased in the PFC of DAT−/− rats. (**D**) ELISA analysis displayed an increase in the PFC tissue levels of IFNγ inflammatory cytokine of DAT−/− rats (myelin basic protein—MBP). Individual data points are shown for each group. Data are presented as mean ± SEM, * *p* < 0.05, ** *p* < 0.01, *** *p* < 0.001.

**Figure 5 biomedicines-09-00157-f005:**
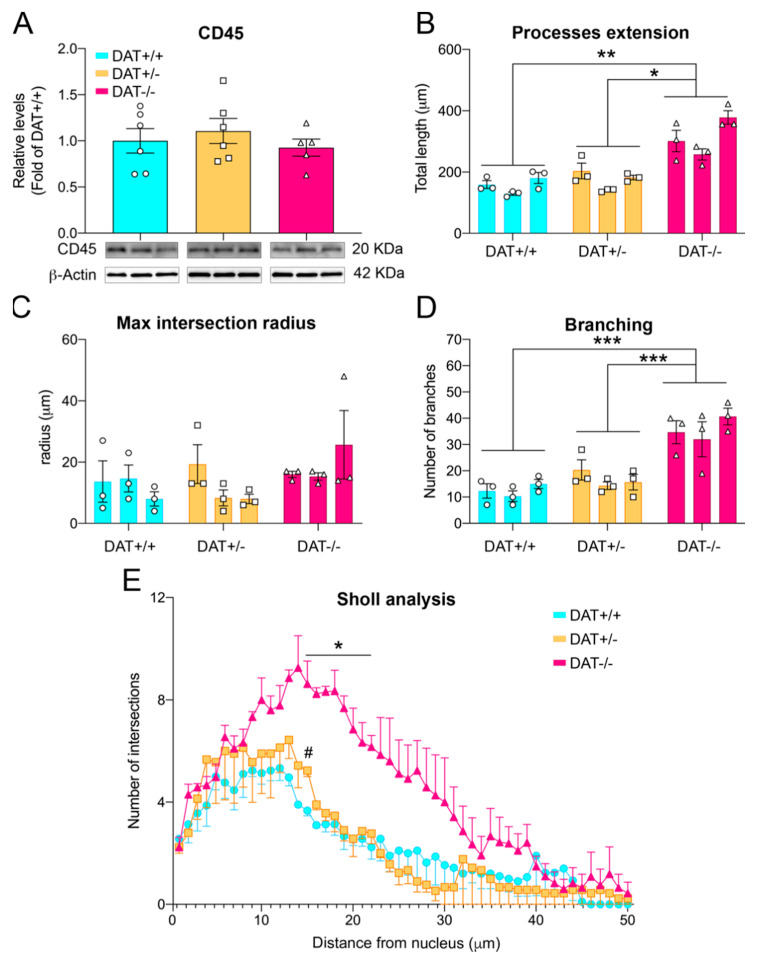
Pro-inflammatory phenotype of microglia in the PFC of DAT−/− rats. (**A**) Levels of CNS microglial marker CD45 were unchanged among the three genotypes. Morphometric analysis of Iba1^+^ cells showed (**B**) an increase in the total length extension of microglial processes of DAT−/− rats, (**C**) while mean maximum radius did not vary among genotypes. (**D**) Number of branches was significantly higher in DAT−/− rats. (**E**) Sholl analysis displayed higher number of intersections in Iba1^+^ cells in DAT−/− PFC. Individual data points are shown for each group. Data are presented as mean ± SEM, * *p* < 0.05, ** *p* < 0.001, *** *p* < 0.001.

**Figure 6 biomedicines-09-00157-f006:**
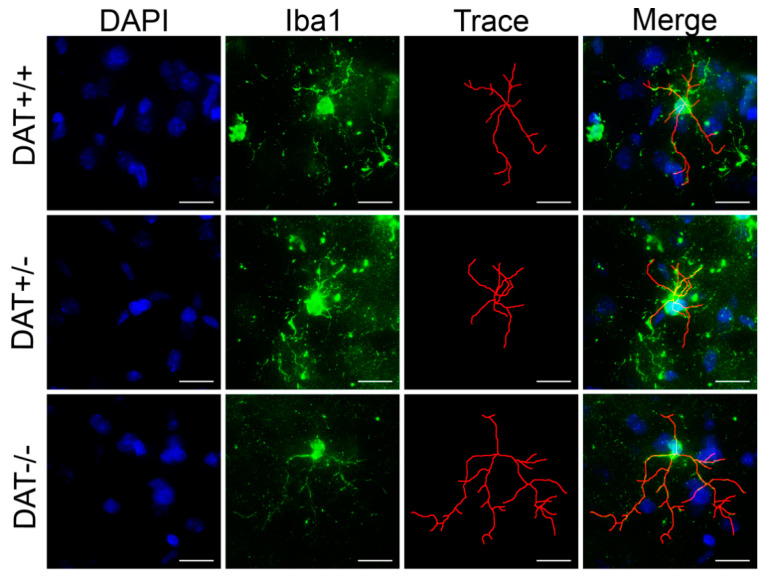
Fluorescence microscopy images display arborization of Iba1+ cells and their 3D morphometric reconstruction in DAT+/+, DAT+/− and DAT−/− rats, highlighting an increase in the number and length extension of microglial branches in the PFC of DAT−/− female rats. Scale bar: 5 μm.

## Data Availability

Data is contained within the article.

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
