# Peer review of "Early Adolescence Prefrontal Cortex Alterations in Female Rats Lacking Dopamine Transporter"

_biomedicines, 2021, doi:10.3390/biomedicines9020157_

Round 1
Reviewer 1 Report
This is the revised manuscript (biomedicines-1081257) by Placido Illano, et al addressing the characterization of the prefrontal cortex function under DAT ablation in adolescent female rats. Authors report that DAT ablation in prefrontal cortex leads to i) hyperactive phenotype, ii) alterations in glutamatergic neurotransmission, iii) signs of neurodegeneration and iV) glial activation. The authors concluded that this study provides new informations about neuroinflammatory pathological processes that occur after DAT ablation in female rats, which in turn can provide novel investigational approaches in human diseases.
The article seems well written and and the results are well documented. It contains interesting information about a matter is very important and noteworthy for the research in the topic of neuropsychiatric diseases. In my opinion the article could be take in consideration for publication after minor revision.
Minor revision:
i) In my opinion, the main limits of the study is the lack of any reference to the different components of dopaminergic pathway. I mean, the authors study the effects of DAT ablation on glutamate receptors, but not, for example, for dopamine receptors. Why?
In my opinion, this aspect would be very important especially in relation to the anti-inflammatory effects that the authors observe in relation to DAT ablation. In this regard, many recent publications underline the role of DA receptors in the regulation of inflammatory processes. The authors could discuss this issue in the discussion sections of the paper.
ii) The authors could better indicate the limits of the study.
iii) Discussion seems long enough work type. I would recommend shortening it.
IV) Check the text for grammatical errors and syntax.
Author Response
This is the revised manuscript (biomedicines-1081257) by Placido Illiano, et al addressing the characterization of the prefrontal cortex function under DAT ablation in adolescent female rats. Authors report that DAT ablation in prefrontal cortex leads to i) hyperactive phenotype, ii) alterations in glutamatergic neurotransmission, iii) signs of neurodegeneration and iV) glial activation. The authors concluded that this study provides new informations about neuroinflammatory pathological processes that occur after DAT ablation in female rats, which in turn can provide novel investigational approaches in human diseases.
The article seems well written and and the results are well documented. It contains interesting information about a matter is very important and noteworthy for the research in the topic of neuropsychiatric diseases. In my opinion the article could be take in consideration for publication after minor revision.
Minor revision:
- i) In my opinion, the main limits of the study is the lack of any reference to the different components of dopaminergic pathway. I mean, the authors study the effects of DAT ablation on glutamate receptors, but not, for example, for dopamine receptors. Why? In my opinion, this aspect would be very important especially in relation to the anti-inflammatory effects that the authors observe in relation to DAT ablation. In this regard, many recent publications underline the role of DA receptors in the regulation of inflammatory processes. The authors could discuss this issue in the discussion sections of the paper.
Thank you for the good observation. We acknowledge the lack of information about DA receptors because is not the main goal of the study. However, we agree on the fact that additional discussion can help the quality of the paper.
Recent studies have demonstrated that dose-dependent activation of dopaminergic receptor operated by dopamine can elicit opposing anti-inflammatory and pro-inflammatory effects, as we now explain in the discussion. We introduced the following test (line 342):
“ Indeed, several studies have confirmed that dopamine can exert either anti-inflammatory and pro-inflammatory effects in human and in-vivo models through dopamine receptors activation on astrocytes and microglia [53-55].
We think that studies addressing the contribution of each of the 5 dopamine receptor in the 2 genotypes of our animal model require accurate studies that might benefit of the use of selective agonist/antagonist, possibly in an ex-vivo electrophysiological recordings settings. At this point, these studies go beyond the scope of our current manuscript, but at the same time provide useful for future in-depth investigations.
- ii) The authors could better indicate the limits of the study.
Thank you for the comment. Starting in line 425, we acknowledge some limitations of the study. We want to highlight that this group of data help the understanding of DAT implication on one important developmental stage and, future studies are needed. As per reviewer suggestion, we have modified and better described additional limitations of the study. We now introduced the following sentence (line 431): “We acknowledge the limitations of the time frame selected and the need of introducing additional time points during development. Future studies…”.
iii) Discussion seems long enough work type. I would recommend shortening it.
We appreciate the suggestion of the reviewer. Based on all the reviewer’s comments and the data presented on the manuscript, we think the discussion is appropriate and its length represents the need to introduce several important aspects to properly discuss, including relevant references, all data presented. If the length of the discussion represents a concern to continue the revision of the manuscript, we will proceed to reduce it. For that, we would really appreciate to know in more detail what information the reviewer thinks is not relevant to discuss the data. We ask the reviewer to please reconsider his/her recommendation.
- iv) Check the text for grammatical errors and syntax.
We thank the reviewer for the review of our manuscript and for the suggestions. We revised the complete manuscript for any grammatical and syntax errors.

Reviewer 2 Report
This study appears at the right time about the need for further understanding DAT in the PFC and in particular when considering sex-differences. The manuscript is well written and enjoyable. Methods are clear and well-detailed.
Here you find some concerns to be addressed and suggestions.
Introduction:
line 63-64: In this section the sentence mentioning astrocytes sounds abrupt. I would suggest to mention any astrocyte involvements before line 56.
Methods:
If all rats were used at PND 35 for any type of experiments, the study should clearly refer anywhere to the age of EARLY adolescence. Indeed, PFC glutamatergic synapses are known to undergo maturational events throughout the adolescence roughly from PND 30 to 75 for female rats. See for example:
- van Zundert et al. Receptor compartmentalization and trafficking at glutamate synapses: a developmental proposal. Trends Neurosci. 2004 doi: 10.1016/j.tins.2004.05.010)
- Willing J, Juraska JM. The timing of neuronal loss across adolescence in the medial prefrontal cortex of male and female rats. Neuroscience. 2015 doi: 10.1016/j.neuroscience.2015.05.073.
- Drzewiecki et al. Synaptic number changes in the medial prefrontal cortex across adolescence in male and female rats: A role for pubertal onset. Synapse. 2016. doi: 10.1002/syn.21909.
Line 131: please specify, if so, that cutting slice were checked for coordinates as referred to the bregma, and which brain atlas you refer to, or otherwise
Results:
Line 233: it is not clear why FJC staining is referred as “preliminary”. It should be either a “not quantitative/example” or “quantitative/analized by statistics”. Please address this issue. Indeed, it sounds redundant studying neuronal cell death by two techniques. Please stress peculiarities derived from using both FJC and NeuN staining, if that is the case.
Discussion
Line 331: Regarding the effect of selective pharmacological inhibition of DAT in the rat PFC, please discuss also in vivo studies such as the recent Biomolecules vol. 10,5 779. 18 May. 2020, doi:10.3390/biom10050779
Line 348: The sentence “Interaction between DA and glutamatergic system is known 52-54”, and relative references, sounds not appropriate. Indeed, the authors should specifically discuss about dopamine’s role in the mesocortical system and the PFC-pyramidal neurons activity (and not in the mesolimbic system, which is involved in diffrent functions).
Line 365: not clear if the sentence “which shows to selectively stain degenerating neurons” is referring to citations 27 and 60 or to presented data. Indeed, authors only presented an example picture as “preliminary”, which sounds not appropriate for a final version manuscript. Please reword.
Author Response
This study appears at the right time about the need for further understanding DAT in the PFC and in particular when considering sex-differences. The manuscript is well written and enjoyable. Methods are clear and well-detailed.
Here you find some concerns to be addressed and suggestions.
Introduction:
- line 63-64: In this section the sentence mentioning astrocytes sounds abrupt. I would suggest to mention any astrocyte involvements before line 56.
We thank the reviewer for the suggestion. We have revised the introduction based on this comment, as follows (line 56):
“Recent studies have also shown the important role played by astrocytes on the control of DA homeostasis in the PFC during early adolescence13,19, when glutamatergic synapses in this area are at an important maturational stage20-22.”
Methods:
- If all rats were used at PND 35 for any type of experiments, the study should clearly refer anywhere to the age of EARLY adolescence.
Thank you for the comment. We have changed “adolescence” for “early adolescence” all across the document. Moreover, we have changed accordingly the text as follows, in line 60:
“In this framework, we sought to investigate the changes that occur in the PFC of female DAT+/- and DAT-/- rats, with a focus on early adolescence”
- Indeed, PFC glutamatergic synapses are known to undergo maturational events throughout the adolescence roughly from PND 30 to 75 for female rats. See for example:
- van Zundert et al. Receptor compartmentalization and trafficking at glutamate synapses: a developmental proposal. Trends Neurosci. 2004 doi: 10.1016/j.tins.2004.05.010)
- Willing J, Juraska JM. The timing of neuronal loss across adolescence in the medial prefrontal cortex of male and female rats. Neuroscience. 2015 doi: 10.1016/j.neuroscience.2015.05.073.
- Drzewiecki et al. Synaptic number changes in the medial prefrontal cortex across adolescence in male and female rats: A role for pubertal onset. Synapse. 2016. doi: 10.1002/syn.21909.
We thank the reviewer for the detailed information provided. We have integrated these references and, following reviewer’s suggestion, we modified the manuscript accordingly (line 56):
“Recent studies have also shown the important role played by astrocytes on the control of DA homeostasis in the PFC during early adolescence13,19, when glutamatergic synapses in this area are at an important maturational stage20-22. “
- Line 131: please specify, if so, that cutting slice were checked for coordinates as referred to the bregma, and which brain atlas you refer to, or otherwise
We thank the reviewer comment and agree that the information is relevant for the methods. We have modified the previous sentence to add the information requested at line 138:
“Coronal serial sections were collected on a freezing microtome (Leica), and coordinates followed were +2.52 mm to −1.44 mm from bregma (antero-posteriorly)26, in 15 series at a thickness of 30 μm”
Results:
- Line 233: it is not clear why FJC staining is referred as “preliminary”. It should be either a “not quantitative/example” or “quantitative/analized by statistics”. Please address this issue.
We thank the reviewer for the accurate comment. According to reviewer’s comment we have modified the results as follows (Line 234):
“In order to assess whether absence of DAT could cause neuronal cell death as previously shown in the DAT-/- mouse model in the striatum31, we initially sought to qualitatively investigate the occurrence of neuronal death by Fluoro-Jade C (FJC) staining32. As shown (Fig. 3A), elevated FJC staining positivity was observed in DAT-/- PFC, while sparse signal was detected in DAT+/-, and complete absence in DAT+/+ controls. Based on FJC qualitative data, we proceeded to investigate marker of apoptotic response such as Cleaved Caspase 333. Western blot analysis showed that levels of Cleaved Caspase 3 higher in the PFC of DAT+/- rats” …. ”In order to further address the occurrence of neuronal cell death and concomitant elevation of Cleaved Caspase 3 expression, we performed PFC NeuN+ cell bodies counting (Fig. 3C, D). PFC tissue from early adolescent females showed a marked reduction in the numbers of NeuN+ cells in DAT-/- genotype [Nested One-way ANOVA F (2, 9) = 10.21; p=0.0048; Tukey’s multiple comparisons test p=0.0048 DAT+/+ versus DAT-/-; p=0.0246 DAT+/- versus DAT-/-].
Taken together, our data revealed neuronal cell death in the PFC of DAT-/- rats.”
- Indeed, it sounds redundant studying neuronal cell death by two techniques. Please stress peculiarities derived from using both FJC and NeuN staining, if that is the case.
We thank the reviewer for the comment. In addressing the presence of neurodegeneration, we proceeded as previously performed by Ehara et al., 2009. In their study, they associated the 2 techniques (FJC staining and NeuN cell count) in order to ascertain the specific loss of neurons in their model of neurodegeneration. Furthermore, we investigated at a whole-tissue whether levels marker of cell death such as cleaved Caspase 3 and phospho P38 were increased. Markers of other glial cell types (CD45 – microglia; MBP – oligodendrocytes) were unchanged and ALDH1L1 astroglial marker was indeed elevated in DAT-/- rats.
After this multi-approach investigation, we believe that we gathered sufficient experimental evidence indicating presence of neurodegeneration (and concurrent neuroinflammation).
We have addressed this in the revised form of the discussion as follows (line 375):
“As previously perfomed32, a dual approach was used to confirm neurodegeneration in DAT-/- PFC, which include i) FluoroJade C (FJC) staining, whose positivity indicates the presence of degenerating neurons32,66, ii) NeuN+ cell bodies count, whose reduction ascertains neuronal loss. We further investigated neurodegeneration addressing the concurrent activation of apoptotic cleaved Caspase 3. Ultimately, we detected no changes in the levels of glial cell type markers CD45 (microglia) and MBP (oligodendrocytes), and increased levels of ALDH1L1 astroglial marker. In conclusion, these data indicate that ablation of DAT is responsible of substantial changes that involve multiple pathways leading to neurodegeneration in the PFC of DAT-/- rats”
Discussion
- Line 331: Regarding the effect of selective pharmacological inhibition of DAT in the rat PFC, please discuss also in vivo studies such as the recent Biomolecules vol. 10,5 779. 18 May. 2020, doi:10.3390/biom10050779
Taking into account the point raised by the reviewer, we have modified the discussion accordingly at line 333:
“Functional state of DAT has been proven to affect neuronal plasticity and control of cognitive function in the PFC50,51, where pharmacological inhibition has shown to induce pro-cognitive effects52.”
- Line 348: The sentence “Interaction between DA and glutamatergic system is known 52-54”, and relative references, sounds not appropriate. Indeed, the authors should specifically discuss about dopamine’s role in the mesocortical system and the PFC-pyramidal neurons activity (and not in the mesolimbic system, which is involved in diffrent functions).
We thank reviewer’s comment. Based on the suggestion of the reviewer we have changed the discussion accordingly (line 354):
“The ensemble of behavioral phenotypes observed in DAT-/- rats10-12,23,24 depends primarily on the different roles that DA neurotransmission plays in the mesocortical system and in the PFC52,58, and it is ascribable to alterations in the cross-talk with glutamatergic neurotransmission59-61. Disruption of these pathways have been widely linked to psychiatric disorders62 including autism, schizophrenia63 and ADHD64. Indeed, adult DAT -/- male rats display glutamate release alterations in the nucleus accumbens54.”
- Line 365: not clear if the sentence “which shows to selectively stain degenerating neurons” is referring to citations 27 and 60 or to presented data.
This part of the discussion has been revised as reported in the answers above (Results – point 2)
- Indeed, authors only presented an example picture as “preliminary”, which sounds not appropriate for a final version manuscript. Please reword.
We appreciate the comment and we kept this consideration in mind when re-writing the results section at line 234 (Reported above in Results – point 1).

Reviewer 3 Report
The manuscript entitled ‘Prefrontal cortex alterations in female adolescent rats lacking dopamine transporter” by Illianio and collaborators, is a descriptive research of behavioral and molecular abnormalities observed in adolescent female DAT-KO rats. The document presents interesting, and potentially very important data of this transgenic model on the alterations of glutamatergic transmission during development. The authors report that the genetic ablation of DAT leads to changes of protein expression of glutamate receptors subunits, altered glial response, increased markers for neuroinflammation, and also found a hyperactive behavioral phenotype in PFC of female rats. Overall, this is an interesting paper with important implications in regards to the dopamine-glutamate interplay in the physiology of the brain during development. Although the study appears to have been carefully conducted, there are a few concerns that must be addressed, and changes that can improve the manuscript.
- The authors make a case for their model being a sex-specific. Although they briefly mentioned in the discussion that they are aware of the missing controls using male rats, they do emphasize in the abstract, introduction and parts of the discussion about how little is known about ‘sex-specific’ alterations that could potentially being associated to psychiatric disorders that are prevalent in females. The authors should tune down such statement. The study is not addressing sex differences because they did not include male rats in the study. At best, they have observed and described changes that occur in female rats, which is very interesting and potentially important if they can find actual sex differences when including male rats in the study and using the appropriate multifactorial statistical model. Seems important to mention as well that circulating hormones modulates synaptic responses. The estrous cycle is an important biological variable that affects both GABAergic and glutamatergic systems and this important variable is not discussed in the manuscript. I would suggest addressing this topic in the discussion.
- In the methods section 2.3. Western Blotting (page three; line 100-103). There is missing information about the centrifugation procedure for the samples. As is described, seems like the authors chose a tissue preparation for protein extracts that is not the best for the proteins they are studying, especially when it comes to presynaptic and postsynaptic proteins. I would be beneficial to see the full, untruncated images for the gels or blots included on the analysis with the molecular marker (as a supplemental figure) or on the response letter to the reviewers. The density of the signals for NMDAR1 and NMDAR2B subunits seems high for the tissue preparation used here, especially when comparing the density found for PSD95. A synaptic fractionation procedure would have been more appropriate for this design. It is understandable however, that using the same tissue samples for several proteins minimized the number of animals utilized, but the specifics about the tissue preparation should be fully disclosed in the methods section, as well as the untruncated images with the molecular weights of the proteins analyzed.
- The authors found differences in some instances for both heterozygous and homozygous rats compared to wild types, whereas differences between heterozygous and homozygous also were reported, but there is no enough information in the discussion about it. I would suggest to add some explanation of why some parameters where affected in both genotypes, and some other were not between genotypes. A justification based on previous work should be added in the discussion section.
- A minor but important detail is not stated in the methods section. The authors refer to PFC, but is not described if they collected both the infralimbic and prelimbic portions of the PFC, or which was the criterion for tissue dissection, and the rationale for why they chose that preparation.
Author Response
The manuscript entitled ‘Prefrontal cortex alterations in female adolescent rats lacking dopamine transporter” by Illianio and collaborators, is a descriptive research of behavioral and molecular abnormalities observed in adolescent female DAT-KO rats. The document presents interesting, and potentially very important data of this transgenic model on the alterations of glutamatergic transmission during development. The authors report that the genetic ablation of DAT leads to changes of protein expression of glutamate receptors subunits, altered glial response, increased markers for neuroinflammation, and also found a hyperactive behavioral phenotype in PFC of female rats. Overall, this is an interesting paper with important implications in regards to the dopamine-glutamate interplay in the physiology of the brain during development. Although the study appears to have been carefully conducted, there are a few concerns that must be addressed, and changes that can improve the manuscript.
- The authors make a case for their model being a sex-specific. Although they briefly mentioned in the discussion that they are aware of the missing controls using male rats, they do emphasize in the abstract, introduction and parts of the discussion about how little is known about ‘sex-specific’ alterations that could potentially being associated to psychiatric disorders that are prevalent in females. The authors should tune down such statement. The study is not addressing sex differences because they did not include male rats in the study. At best, they have observed and described changes that occur in female rats, which is very interesting and potentially important if they can find actual sex differences when including male rats in the study and using the appropriate multifactorial statistical model.
We thank the reviewer for the comment. In line with this, we have reframed the abstract, introduction and discussion accordingly. Changes have been made across all document (for example, “Sex-specific” has been removed from abstract, first paragraph of introduction, and in third paragraph of discussion).
Seems important to mention as well that circulating hormones modulates synaptic responses. The estrous cycle is an important biological variable that affects both GABAergic and glutamatergic systems and this important variable is not discussed in the manuscript. I would suggest addressing this topic in the discussion.
We really appreciate the reviewer for the comment. We agree that circulating hormones play an important role on synaptic responses. However, we chose PND 35 for our study, time point that has been described as before the first proestrus appears. As Ajayi discuss in his recent publication (Ajayi and Akhigbem 2020), “In female rats, puberty is preceded by the pulsatile release of luteinizing hormone (LH) after the 4th postnatal week, approximately 30 days old [Ekambaram et al., 2017]. This period is the anestrus and occurs about 8 to 9 days before the first proestrus [Foitzik et al., 2000]. The first proestrus, estrus, metestrus and diestrus then follow.” We anticipate not to have relevant hormone effects on the data presented. However, we acknowledge the relevant point that the reviewer does and, accordingly, we have added a sentence on the discussion (Line 427) as follows:
“Additionally, we studied female rats on PND35, time point during development when hormones have been described to be almost ready to start its first proestrus cycle80. We acknowledge the need of future studies considering the role of hormones modulating glutamatergic synaptic response.”
Ekambaram G, SKS K, Joseph LD. Comparative Study on the Estimation of Estrous Cycle in Mice by Visual and Vaginal Lavage Method. J Clin Diagnostic Res. 2017;11(1):AC05–AC07.
Foitzik T, Hotz HG, Eibl G, Buhr HJ. Experimental models of acute pancreatitis: are they suitable for evaluating therapy? Int J Color Dis. 2000;15:127–135. doi: 10.1007/s003840000216.
Ajayi AF, Akhigbe RE. Staging of the estrous cycle and induction of estrus in experimental rodents: an update. Fertil Res Pract. 2020;6:5. Published 2020 Mar 14. doi:10.1186/s40738-020-00074-3
- In the methods section 2.3. Western Blotting (page three; line 100-103). There is missing information about the centrifugation procedure for the samples. As is described, seems like the authors chose a tissue preparation for protein extracts that is not the best for the proteins they are studying, especially when it comes to presynaptic and postsynaptic proteins. I would be beneficial to see the full, untruncated images for the gels or blots included on the analysis with the molecular marker (as a supplemental figure) or on the response letter to the reviewers.
The density of the signals for NMDAR1 and NMDAR2B subunits seems high for the tissue preparation used here, especially when comparing the density found for PSD95. A synaptic fractionation procedure would have been more appropriate for this design. It is understandable however, that using the same tissue samples for several proteins minimized the number of animals utilized, but the specifics about the tissue preparation should be fully disclosed in the methods section, as well as the untruncated images with the molecular weights of the proteins analyzed.
We are appreciative of the feedback; we acknowledge the improvement on the methods following reviewer’s comment. According to reviewer’s comment we have added the following sentence on the text where we have extended the information in regards to the preparation of western blot samples (Line 109):
“ Samples were left on ice for 20 minutes and the tip-sonicated for 3 seconds on medium speed. Samples were centrifuged for 20 minutes at 4 °C at 10000 g”. Indeed, this technique allowed us to fully implement the 3R principles, and to study membrane protein, cytoplasm proteins and also to carry out ELISA experiments, using tissue samples for the very same rats used for behavioral analysis. All raw western gel pictures are now included as supplementary material.
- The authors found differences in some instances for both heterozygous and homozygous rats compared to wild types, whereas differences between heterozygous and homozygous also were reported, but there is no enough information in the discussion about it. I would suggest to add some explanation of why some parameters where affected in both genotypes, and some other were not between genotypes. A justification based on previous work should be added in the discussion section.
Based on reviewer’s comment, we added a paragraph in the discussion addressing this matter (line 414).
“Our differential data obtained from DAT-/- and DAT+/- rats might, therefore, provide further insights in the PFC pathophysiology of neuropsychiatric diseases like ADHD and PTSD respectively, whose model validity was demonstrated in our previous work9. Increasing number of studies using DAT+/- and DAT-/- rats are on their way, and based on our results and previous publications10,12, partial mutation on DAT is sufficient to cause several alterations on a variety of underlying mechanisms controlling, between other steps, behavioral parameters relevant for the study of psychiatric disorders9,10,12.
- A minor but important detail is not stated in the methods section. The authors refer to PFC, but is not described if they collected both the infralimbic and prelimbic portions of the PFC, or which was the criterion for tissue dissection, and the rationale for why they chose that preparation.
Prelimbic portion of PFC was dissected as now reported in the Methods, line 105. The rational of this choice is now discussed in the main manuscript, line 344: “In this framework, we sought to investigate the effects of DAT ablation in the prelimbic PFC, responsible for cognitive processing64 that could lead to glial pro-inflammatory process that might contribute to the underlying pathophysiology of female DAT+/- and DAT -/- rats during early adolescence.”

Reviewer 4 Report
The present study aims at better understanding the neuroinflammatory processes that occur during adolescence in rats lacking, fully or partially, the dopamine transporter. To this purpose, the authors carry out behavioral, glutamatergic, neurodegenerative and glial investigations in homozygous and heterozygous DAT knock out female rats. Overall the paper is easy to read, figures are nice, and results and material and methods are well-written and clear. However, the scientific impact of the study is limited and could be improved. Indeed, the rationale supporting the general purpose lacks consistency in the introduction, and results are not fully exploited in the discussion. General and minor comments are reported below.
Major comments :
- Based on the introduction, the aim of the present study is to investigate « the developmental changes that occur in the DAT-KO rat model focusing on adolescent female rats ». To this purpose, the authors plan to assess different parameters such as locomotor activity, glutamate receptor expression, cell death, glial markers…However, the developmental aspect of the study is actually beyond the study, as the only stage which is investigated in the present paper is adolescence. Also, no rationale is provided to support the investigations on the glutamate system in the introduction. On the other hand, the authors precise in the discussion that the aim is to « investigate the involvement of glial cells in the underlying pathophysiology of female DAT-KO rats during adolescence ». This is confusing, the working hypothesis should be clear from the start to the end of the manuscript, it should strictly reflect the data of the paper, and all paramaters studied should be justified by a proper rationale.
- Methods : The authors used het mothers for their breeding scheme. Did they check that maternal behavior was not affected by this genotype ? If there exist some references in the literature discarding such an effect or any impact of a change in DAT-related maternal behavior on pups development and behavior at adolescence, it should be cited.
- Discussion, lines 345 and 412: the authors state that their data provide a sex- and age-specific occurrence of sustained hyperactive phenotype in adolescent female rats. These are over-statements, as they did not carry out their experiments in males and at other developmental stages to establish a sex- and age-specific observation.
- Discussion : It is somehow strange, for a paper that focuses on the effect of the deletion of the dopamine transporter, that dopamine is never included itself in the discussion. In this regard, no corroboration between data from locomotor activity experiments and the other investigations is ever provided. Also, the authors never discuss a possible gene-dosage effect by systematically corroborating data obtained from their experiments in homozygous and heterozygous KO rats. Finally, it is clear that the authors wish to present here a sex-specific effec of DAT deletion, although, as they write, they cannot state on this matter considering that the study was ran only in females. However, it would be welcome to compare the data obtained from the literature with the ones obtained in the present paper to discuss this possible sex-specific effect.
Minor comments :
- Abstract, lines 22-23, « Currently, treatment of these pathologies still represents a challenge, and it exists a gap in sex-specific knowledge of these disorders also addressing development » : this sentence is not clear (especially the end), please rephrase.
- Abstract : no clear hypothesis or rationale is provided. The authors should improve the abstract in this regard.
- Introduction, line 41 : please replace « genotype » with « genotypes »
- Introduction, line 42 : please replace « form » with « from »
- Introduction, lines 40-42, « adult dopamine transporter knockout (DAT-KO) rats, in the heterozygous and homozygote genotype » : This is confusing, as throughout the text KO is used for the homozygous genotype. Please prefer dopamine transporter deletion, or use homozygous or heterozygous KO in the whole manuscript. Also, the authors should be more specific about the genotype (homozygous, heterozygous) which displays ADHD in males (line 43).
- Introduction, lines 45-47, « These findings render this animal model highly suitable for sex-specific investigations during development in the pathophysiology of disorders involving aberrant DA function related to DAT ablation » : it seems that the authors plan to run investigations both in males and females. The listed data from the literature justifies the use of the model to study DAT dependent alterations but not the sex-specific impact of DAT deletion.
- Introduction, lines 60-62 : It seems strange to re-introduce the PFC here, while the focus on this brain region has already been justified by the previous paragraph.
- Introduction, line 69 : The authors do not mention females, while this is the core of the present paper.
- Methods, line 127 : The tissue collection is already explained in the previous paragraph. Methods would be improved by taking this out and precising in « western blotting » section what was the other half brain used for (or preparing a new paragraph dedicated to tissue collection).
- Methods, line 167 : please replace « inteferon » with « interferon »
- Results, lines 184-186, « 3.1. Locomotor activity of female adolescent DAT-KO rats » : the first part of this paragraph is redundant with the Methods section.
- Figure 1D : The variability of the homozygous DAT KO group is very high, compared to the other paramaters assessed in this experiment. Did the authors check whether the data from this group include different populations with high and low vertical activity counts?
- Figure 3A : It would be nice to see the quantification graph for fluoro jade C levels.
- Results, lines 241-242 : A verb is missing at the beginning of the sentence (or « whether » should be taken out).
- Results, line 242 : please replace « numbers » with « number » and « cells bodies » with « cell bodies »
- Results, line 265 : please replace « oligodendorcytes » with « oligodendrocytes »
- Results, line 289 : please replace « marker » with « a marker »
- Discussion : It is strange that the authors decided to mention DTDS for the very first time in the statement paragraph of the discussion. I would suggest to include it in the introduction or take it out.
- Discussion, line 342 : please replace « indicates » with « indicate »
- Discussion, line 365 : « which shows to selectively stain degenerating neurons », not clear, please rephrase
- Discussion, line 371 : « at specifically address changes in this genotype », not clear, please rephrase.
- Discussion, lines 347-359 : The authors should try to speculate on the mechanisms responsible for the impact of DAT KO on the changes in glutamate receptor expression.
- Discussion, line 390 : please replace « regulate » with « regulates »
Author Response
The present study aims at better understanding the neuroinflammatory processes that occur during adolescence in rats lacking, fully or partially, the dopamine transporter. To this purpose, the authors carry out behavioral, glutamatergic, neurodegenerative and glial investigations in homozygous and heterozygous DAT knock out female rats. Overall the paper is easy to read, figures are nice, and results and material and methods are well-written and clear. However, the scientific impact of the study is limited and could be improved. Indeed, the rationale supporting the general purpose lacks consistency in the introduction, and results are not fully exploited in the discussion. General and minor comments are reported below.
Major comments :
- Based on the introduction, the aim of the present study is to investigate « the developmental changes that occur in the DAT-KO rat model focusing on adolescent female rats ». To this purpose, the authors plan to assess different parameters such as locomotor activity, glutamate receptor expression, cell death, glial markers…However, the developmental aspect of the study is actually beyond the study, as the only stage which is investigated in the present paper is adolescence.
We thank the reviewer for his comment. We have modified the cite about the aim of the study to be more descriptive of the time frame we cover. Following reviewer’s suggestion, we have modified the following sentence as shown (line 60):
“In this framework, we sought to investigate the changes that occur in the PFC of female DAT+/- and DAT-/- rats, with a focus on early adolescence“.
- Also, no rationale is provided to support the investigations on the glutamate system in the introduction.
We really appreciate the comment. We agree that the introduction missed citation of the purpose of the glutamate study. As per reviewer’s suggestion we have added the following (Line 61):
“We aimed to provide a more comprehensive understanding of the role of DAT dysfunction and its impact on glial homoeostasis, for its role on the pathophysiology of several psychiatric and neurodegenerative diseases,. Furthermore, we also investigated parts of the glutamatergic signaling due its known interaction with the DA system23-25, and dysfunctions described in a variety of psychiatric disorders26-28 with high prevalence in women and onset during adolescence.“
- On the other hand, the authors precise in the discussion that the aim is to « investigate the involvement of glial cells in the underlying pathophysiology of female DAT-KO rats during adolescence ». This is confusing, the working hypothesis should be clear from the start to the end of the manuscript, it should strictly reflect the data of the paper, and all paramaters studied should be justified by a proper rationale.
Thank you for the precise comment. We agree with the reviewer’s suggestion and we have changed the discussion accordingly (Line 344) to reflect the specific data we present:
“In this framework, we sought to investigate the effects of DAT ablation in the prelimbic PFC, responsible for cognitive processing64 that underlie the behavioral phenotype of DAT-/- rats, and that could lead to glial pro-inflammatory process that might contribute to the underlying pathophysiology of female DAT+/- and DAT -/- rats during early adolescence.”
- Methods : The authors used het mothers for their breeding scheme. Did they check that maternal behavior was not affected by this genotype? If there exist some references in the literature discarding such an effect or any impact of a change in DAT-related maternal behavior on pups development and behavior at adolescence, it should be cited.
We appreciate the important point raised here by the reviewer. Most of the studies performed with DAT rat colony, from our group as well as others, has been done following HET-HET (DAT+/- female paired with DAT+/- male) breeding9-12,29. Our main goal is to characterize these rats following previous studies, to provide additional information about their phenotype to allow their use for future studies with focus on a variety of psychiatric disorders. We acknowledge the limitations that this could have on the results, however, is out of our scope of work at this point. To answer the reviewer, yes, even though the majority of studies performed with DAT rats use same breeding strategy, our group recently reported (30) the impact that different breeding strategies can have on the behavioral outcome of DAT KO rats. We agree that further studies should have into consideration when interpreting the data, the breeding strategy chosen.
As per reviewer suggestion, we have added the following sentence on the methods (line 82): “We acknowledge the possible effect that the breeding strategy could have on the study30, out of our scope of work.“
- Discussion, lines 345 and 412: the authors state that their data provide a sex- and age-specific occurrence of sustained hyperactive phenotype in adolescent female rats. These are over-statements, as they did not carry out their experiments in males and at other developmental stages to establish a sex- and age-specific observation.
We thank the reviewer’s comment. Following reviewer’s advice, we have removed the sentence in previous line 412 (DAT-KO rat model represents a highly suitable model for sex-specific investigations in the pathophysiology of disorders involving aberrant function of DAT).
- Discussion: It is somehow strange, for a paper that focuses on the effect of the deletion of the dopamine transporter, that dopamine is never included itself in the discussion. In this regard, no corroboration between data from locomotor activity experiments and the other investigations is ever provided.
Following reviewer’s comment, we have added several references in regard to DA in the discussion. Please see the following sentence that has been added into the discussion (Line 342):
“Indeed, several studies have confirmed that DA can exert either anti-inflammatory and pro-inflammatory effects in human and in-vivo models through dopamine receptors activation on astrocytes and microglia61-63.”
We do, however, introduce results from previous literature regarding the locomotor activity; please see line 349:
“Our data indicate that adolescent DAT-/- female rats display an overt hyperactive phenotype in the locomotor chamber, characterized by increased sustained locomotion as well as stereotypic behavior. Our data confirm previous findings observed in this animal model9,10,29 and demonstrate occurrence of sustained hyperactive phenotype in adolescent female rats.”
- Also, the authors never discuss a possible gene-dosage effect by systematically corroborating data obtained from their experiments in homozygous and heterozygous KO rats.
We would like to thank reviewer’s observation. Some reviewers are concerned about the extension of the discussion. However, we briefly mentioned on the discussion the following (line 383):
“Interestingly, a trend in increase of cleaved Caspase 3 and spare positivity to FJC staining was observed in female DAT+/- rats, that, in combination with the reduced NMDAR2B protein levels, open novel investigational scenarios oriented at specifically address changes that could be occurring when a mutation is only present in one allele and DAT is only partially reduced”.
We also point out in the discussion that (line 414):
“Our differential data obtained from DAT-/- and DAT+/- rats might, therefore, provide further insights in the PFC pathophysiology of neuropsychiatric diseases like ADHD and PTSD respectively, whose model validity was demonstrated in our previous work9.”
- Finally, it is clear that the authors wish to present here a sex-specific effect of DAT deletion, although, as they write, they cannot state on this matter considering that the study was ran only in females. However, it would be welcome to compare the data obtained from the literature with the ones obtained in the present paper to discuss this possible sex-specific effect.
We acknowledge the need to study male DAT rats for such statements. We indeed present the limitation of our work (line 424):
“Although most psychiatric disorders cited across our study present higher prevalence in women, we acknowledge that our study focused only on female rats and further studies are needed to expand our results also in male rats. The present investigation focused on early adolescent rats.”
However, across literature during last decade, there is an increasing willingness to specifically cite female results. However, is still common to find male and female data combined to reduce the number of animals when the main focus of the study is not sex differences. Our study focused on female rats. We do not exclude the need of the study of same parameters in male rats. We acknowledge the need of further studies with this purpose as shown above.
Following reviewer’s comment, we have removed sentence on initially line 418: “DAT-KO rat model represents a highly suitable model for sex-specific investigations in the pathophysiology of disorders involving aberrant function of DAT”. Additionally, we have also removed in line 426 “in a sex-specific manner”.
Minor comments :
- Abstract, lines 22-23, Currently, treatment of these pathologies still represents a challenge, and it exists a gap in sex-specific knowledge of these disorders also addressing development » : this sentence is not clear (especially the end), please rephrase.
Sentence has been rephrased as follows: “Currently, a gap in the knowledge of these disorders in adolescent females still represents a major limit for the development of appropriate treatments.”
- Abstract : no clear hypothesis or rationale is provided. The authors should improve the abstract in this regard.
Sentence has been changed/added: “The present work focuses on the characterization of the PFC function under condition of heterozygous and homozygous ablation of DAT during early adolescence based on the known implication of DAT and PFC DA in psychopathology during adolescence.”
- Introduction, line 41 : please replace « genotype » with « genotypes »
It has been replaced.
- Introduction, line 42 : please replace « form » with « from »
It has been replaced
- Introduction, lines 40-42, « adult dopamine transporter knockout (DAT-KO) rats, in the heterozygous and homozygote genotype » : This is confusing, as throughout the text KO is used for the homozygous genotype. Please prefer dopamine transporter deletion, or use homozygous or heterozygous KO in the whole manuscript.
Sentence “adult dopamine transporter knockout (DAT-KO) rats, in the heterozygous and homozygote genotype”) has been changed as suggested. We modified it as follows “Recent studies focusing on adult dopamine transporter deletion in rats, in the heterozygous and homozygote genotypes….,”.
Per reviewer suggestion, we have changed all across the manuscript DAT-KO for DAT-/- and DAT+/- where pertinent, to be consistent and make the reading/understanding easier.
- Also, the authors should be more specific about the genotype (homozygous, heterozygous) which displays ADHD in males (line 43).
Both genotypes (DAT+/- and DAT-/-) present phenotypic aberrations that could make both valid models for ADHD (Please see reference 10).
- Introduction, lines 45-47, « These findings render this animal model highly suitable for sex-specific investigations during development in the pathophysiology of disorders involving aberrant DA function related to DAT ablation »: it seems that the authors plan to run investigations both in males and females. The listed data from the literature justifies the use of the model to study DAT dependent alterations but not the sex-specific impact of DAT deletion.
Due to the lack of data with separated groups of males and females, we agree with the reviewer. The sentence has been modified as follows: “In line with our previous studies, DAT-/- female rats provide a highly suitable animal model to investigate the less known pathophysiology of disorders involving aberrant DA function related to DAT ablation affecting females.”
- Introduction, lines 60-62 : It seems strange to re-introduce the PFC here, while the focus on this brain region has already been justified by the previous paragraph.
Sentence in line 60 has been moved to the end of the previous paragraph.
- Introduction, line 69: The authors do not mention females, while this is the core of the present paper.
Additional information has been added in line 64 as follows:
“Furthermore, we also investigated parts of the glutamatergic signaling due its known interaction with the DA system23-25, and dysfunctions described in a variety of psychiatric disorders26-28 with high prevalence in women and onset during adolescence.”
- Methods, line 127: The tissue collection is already explained in the previous paragraph. Methods would be improved by taking this out and precising in « western blotting » section what was the other half brain used for (or preparing a new paragraph dedicated to tissue collection).
As suggested, the sentence has been changed as follows (line 134): “Rats were sacrificed as indicated in the “western blotting” section”.
- Methods, line 167 : please replace « inteferon » with « interferon »
It has been changed
- Results, lines 184-186, « 3.1. Locomotor activity of female adolescent DAT-KO rats » : the first part of this paragraph is redundant with the Methods section.
As suggested, first part of the paragraph has been removed.
- Figure 1D : The variability of the homozygous DAT KO group is very high, compared to the other paramaters assessed in this experiment. Did the authors check whether the data from this group include different populations with high and low vertical activity counts?
Thank you for the comment. DAT -/- rats present high levels of hyperactivity in a variety of parameters, including stereotypic behavior. They tend to run obsessively across the walls. However, there is more variability across the animals but no statistical standard deviation criteria was met to exclude some rats. There are rats with high, medium and low vertical activity.
- Figure 3A : It would be nice to see the quantification graph for fluoro jade C levels.
The FJC staining was done only in 1-2 rats per group as qualitative to guide the next NeuN staining study. As stated in the text, it is only preliminary so no quantification can be done.
- Results, lines 241-242 : A verb is missing at the beginning of the sentence (or « whether » should be taken out).
The sentence has been modified as follows (Line 246): “In order to further address the occurrence of neuronal cell death and concomitant elevation of Cleaved Caspase 3 expression…”
- Results, line 242 : please replace « numbers » with « number » and « cells bodies » with « cell bodies »
Both have been replaced.
- Results, line 265 : please replace « oligodendorcytes » with « oligodendrocytes »
It has been replaced.
- Results, line 289 : please replace « marker » with « a marker »
It has been added.
- Discussion : It is strange that the authors decided to mention DTDS for the very first time in the statement paragraph of the discussion. I would suggest to include it in the introduction or take it out.
We have removed the extensive explanation about DTSD on the discussion
- Discussion, line 342 : please replace « indicates » with « indicate »
It has been replaced.
- Discussion, line 365 : « which shows to selectively stain degenerating neurons », not clear, please rephrase
It has been rephrased as follows (line 376): “FluoroJade C (FJC) staining, whose positivity indicates the presence of degenerating neurons28,64…”
- Discussion, line 371 : « at specifically address changes in this genotype », not clear, please rephrase.
The sentence has been rephrased as follows (line 386): “….scenarios oriented at specifically address changes that could be occurring when a mutation is only present in one allele and DAT is only partially reduced, in DAT+/- rats”.
- Discussion, lines 347-359 : The authors should try to speculate on the mechanisms responsible for the impact of DAT KO on the changes in glutamate receptor expression.
We would like to thank reviewer’s suggestion. Some reviewers are concerned about the extension of the discussion. However, per reviewer’s comment, we have added the following (Line 367): “”. Further studies investigating the mechanisms responsible for such impact on glutamatergic signaling are needed. However, the hyperdopaminergic state on DAT-/+ as well as DAT-/- could be underlying such impact on the glutamatergic pathway components.
- Discussion, line 390 : please replace « regulate » with « regulates »
It has been replaced.

Round 2
Reviewer 3 Report
I appreciate the consideration taken to my comments. The authors have satisfactorly addressed my concerns.